# Specificity of Anti-Citrullinated Protein Antibodies to Citrullinated α-Enolase Peptides as a Function of Epitope Structure and Composition

**DOI:** 10.3390/antib10030027

**Published:** 2021-07-21

**Authors:** Ilaria Fanelli, Paolo Rovero, Paul Robert Hansen, Jette Frederiksen, Gunnar Houen, Nicole Hartwig Trier

**Affiliations:** 1Interdepartmental Laboratory of Peptide and Protein Chemistry and Biology, Department of NeuroFarBa, University of Florence, 50019 Sesto Fiorentino, Italy; ilaria.fanelli1@stud.unifi.it (I.F.); paolo.rovero@unifi.it (P.R.); 2Department of Drug Design and Pharmacology, University of Copenhagen, 2100 Copenhagen, Denmark; prh@sund.ku.dk; 3Department of Neurology, Rigshospitalet Glostrup, 2600 Glostrup, Denmark; jette.lautrup.battistini@regionh.dk (J.F.); gunnar.houen@regionh.dk (G.H.); 4Department of Biochemistry and Molecular Biology, University of Southern Denmark, 5230 Odense, Denmark

**Keywords:** anti-citrullinated protein antibodies, citrullinated peptides, epitopes, rheumatoid arthritis

## Abstract

Rheumatoid arthritis (RA) is an autoimmune disease affecting approximately 1–2% of the world population. In addition to the first discovered serologic markers for RA, the rheumatoid factors (RFs), anti-citrullinated protein antibodies (ACPAs) are even more specific for the disease compared to RFs and are found in 70–80% of RA patient sera. RA etiopathogenesis still needs to be elucidated, as different factors are proposed to be involved, such as Epstein–Barr virus infection. Hence, understanding the interaction between ACPAs and their citrullinated peptide targets is relevant for a better knowledge of RA pathophysiology and for diagnostic purposes. In this study, a cohort of RA sera, healthy control sera and multiple sclerosis sera were screened for reactivity to a variety of citrullinated peptides originating from α-enolase, pro-filaggrin, proteoglycan and Epstein–Barr nuclear antigen-2 by enzyme-linked immunosorbent assay. ACPA reactivity to citrullinated α-enolase peptides was found to depend on peptide length and peptide conformation, favouring cyclic (disulfide bond) conformations for long peptides and linear peptides for truncated ones. Additional investigations about the optimal peptide conformation for ACPA detection, employing pro-filaggrin and EBNA-2 peptides, confirmed these findings, indicating a positive effect of cyclization of longer peptides of approximately 20 amino acids. Moreover, screening of the citrullinated peptides confirmed that ACPAs can be divided into two groups based on their reactivity. Approximately 90% of RA sera recognize several peptide targets, being defined as cross-reactive or overlapping reactivities, and whose reactivity to the citrullinated peptide is considered primarily to be backbone-dependent. In contrast, approximately 10% recognize a single target and are defined as nonoverlapping, primarily depending on the specific amino acid side-chains in the epitope for a stable interaction. Collectively, this study contributed to characterize epitope composition and structure for optimal ACPA reactivity and to obtain further knowledge about the cross-reactive nature of ACPAs.

## 1. Introduction

RA is a systemic and chronic autoimmune disease with a worldwide prevalence of approximately 5 per 1000 adults, affecting women two to three times more often than men. RA disease onset may occur at any age; however, the peak incidence is in the sixth decade [1,2,3]. RA is characterized by infiltration of monocytes, B cells and T cells in the synovial membrane in joints and ultimately cartilage degradation and erosion of the underlying bone [2]. In addition to joint damage, some systemic features are associated with RA, for instance pulmonary, cardiovascular, psychological, and skeletal disorders [4,5]. Hence, RA has increased morbidity and mortality rates when left untreated [1,6].

RA is diagnosed according to EULAR/ACR classification criteria revised in 2010 which, along with clinical disease manifestations, comprise serological biomarkers such as anti-citrullinated protein antibodies (ACPA) [7]. ACPAs are detected in 70–80% of RA patients and in approximately 1–2% of the healthy population. Moreover, they have been reported to be present up to 14 years before the manifestation of clinical symptoms, making ACPAs good biomarkers for RA [8,9]. Ultimately, it has been reported that ACPA-positive RA patients experience increased joint damage and low remission rates, indicating that these individuals have more severe disease courses compared to ACPA-negative RA patients [10]. The occurrence of ACPA-positive RA is related to genetic risk factors that predispose for RA, for instance, protein tyrosine phosphatase nonreceptor type-22 (PTPN22) and MHC class II alleles [11,12,13,14].

ACPAs recognize the nonstandard amino acid citrulline, a nongenetically encoded amino acid. Citrullination is the result of a post-translational modification, where the positively charged guanidino group of Arg is substituted by the neutral ureido group. Ultimately, this modification may lead to structural unfolding of the citrullinated protein [15,16]. Citrullination is catalyzed by Peptidyl Arginine Deiminase (PAD) enzymes, which are calcium-dependent metalloenzymes [17].

ACPAs are typically detected with assays, which exploit enzyme-linked immunosorbent assay (ELISA) methods and synthetic citrullinated peptides [18,19,20,21,22]. The first generation of assays was based on a synthetic linear citrullinated peptide derived from human filaggrin [19]. In order to improve assay sensitivity, the linear peptide was replaced by a cyclic version (Cyclic Citrullinated Peptide, CCP, containing a disulfide bond), as the cyclic peptide yielded higher sensitivity and specificity compared to the linear version. This assay is also referred to as CCP1 [20]. Screening of peptide libraries has led to the selection of other antigens and generation of second and third generations of CCP assays [18]. While the previously mentioned assays only detect IgG ACPAs, the CCP3.1 detects both IgG and IgA isotypes. Despite this, the golden standard for ACPAs detection is the CCP2 assay [18,22].

In the commercial ACPA assays, different citrullinated peptides are employed, which is in accordance with the cross-reactive nature of ACPAs. ACPAs are able to recognize several citrullinated targets, preferably containing a Cit–Gly motif [19,21,23,24,25,26,27]. Besides a critical Cit–Gly motif, charged amino acids in the C-terminal have been proposed to be essential for a stable interaction between ACPAs and citrullinated peptide targets [23]. The amino acids surrounding citrulline have been analyzed in several studies, which revealed that substitutions in positions -x-x-Cit-Gly-x- do not influence antibody reactivity. This finding demonstrates the crucial role of the Cit–Gly motif for a stable antibody–antigen interaction, even though sometimes other amino acids besides Gly are also tolerated [23,24,26,27,28]. Examples of ACPA targets that have been reported are collagen, fibrinogen, α-enolase, vimentin, pro-filaggrin, Epstein–Barr nuclear antigen (EBNA)-1, and EBNA-2 [21,22,25].

It has been proposed that ACPAs can be divided into two groups, based on their ability to interact with citrullinated peptides [16,17], one group that appears to recognize a large variety of citrullinated targets and another group that recognizes a very limited number of citrullinated peptides [29]. The first group of ACPAs, also referred to as “overlapping” or “cross-reactive” antibodies, is primarily backbone-dependent, whereas the second group, also referred to as “nonoverlapping” or “epitope-specific”, depends on the specific amino acid side-chains of the epitope to establish a stable antibody–antigen interaction [16].

On this basis, we analyzed the interactions between citrullinated targets and ACPAs in order to obtain further knowledge about ACPAs, which is important for the improvement of diagnostic tools, and to elucidate their role in the pathogenesis of RA. Citrullinated α-enolase peptides were used as a point of origin to characterize epitope composition and structure for optimal antibody reactivity and the overlapping and nonoverlapping ACPA reactivities.

## 2. Materials and Methods

### 2.1. Reagents

Alkaline phosphatase (AP)-conjugated goat-anti-human IgG, streptavidin and AP substrate tablets (*para*-nitrophenylphosphate (*p*NPP)) were from Sigma Aldrich (St. Louis, MO, USA). Tris-Tween-NaCl (TTN, 0.3 M NaCl, 20 mM Tris, 0.01% Tween 20, pH 7.5), carbonate buffer (0.05 M sodium carbonate, pH 9.6) and AP substrate buffer (1 M ethanolamine, 0.5 mM MgCl_2_, pH 9.8), were from Statens Serum Institut (Copenhagen, Denmark). Synthetic peptides purchased were from Schäfer-N (Lyngby, Denmark) (Table 1) and were generated on TentaGel resin using standard Fmoc-based solid-phase peptide synthesis. The peptides were synthesized as peptide acids.

### 2.2. Patient Material

RA serum samples (*n* = 28) and healthy donor (*n* = 28) serum samples (referred to as healthy control (HC)) were provided by Statens Serum Institut Biobank (Copenhagen, Denmark) (*n* = 28), which routinely analyzes patient sera for diagnostic purposes. The samples were tested anonymously, therefore not requiring ethical consent.

Ten multiple sclerosis serum samples from the Multiple Sclerosis Clinic, Department of Neurology, Rigshospitalet Glostrup (Glostrup, Denmark) were used as disease controls. The samples were tested anonymously, therefore not requiring ethical consent.

### 2.3. Detection of Antibodies by Enzyme-Linked Immunosorbent Assay and Streptavidin-Capture Enzyme-Linked Immunosorbent Assay

Microtiter plates were coated with 1 µg/mL free peptide in carbonate buffer and incubated overnight at room temperature (RT) on a shaking table (ST). The wells were rinsed with TTN for 3 × 1 min and blocked with TTN for 30 min. Sera were diluted (1:200) in TTN, added to each well, and then incubated for 1 h (h) at RT on a ST. After washing with TTN buffer, AP-conjugated goat-anti-human IgG diluted in TTN (1:1000) was added to each well and incubated for 1h at RT on a ST. Finally, *p*NPP-containing AP substrate buffer (1 mg/mL) was added to each well and AP activity was determined by measuring the absorbance at 405 nm with background subtraction at 650 nm.

Alternatively, microtiter plates were precoated with 1 µg/mL streptavidin in carbonate buffer and incubated overnight at 4 °C. Biotinylated peptides (diluted to 1 µg/mL in carbonate buffer) were added to each well and incubated for 2h at RT on a ST. The following steps in the experiment were carried out as mentioned above. All samples were tested in duplicates.

Based on preliminary screening, absorbances of all the results were normalized to a positive RA control pool (*n* = 28) and a peptide-specific cutoff was introduced, tolerating a nonspecific reactivity of 5% and an intra-assay variation of 15%. Readings above the cutoff were regarded as positive, whereas samples below the cutoff were regarded as being negative. Inter-assay variations below 15% were acceptable.

### 2.4. Statistics

Statistical analyses and plots were generated using GraphPad Prism 9.0 software. The values obtained in the experiments were compared further by using Student’s *t*-test.

## 3. Results

### 3.1. Reactivity of Rheumatoid Arthritis Sera to α-Enolase Peptides

Various citrullinated protein targets have been identified in RA, such as α-enolase, pro-filaggrin, proteoglycan, and fibronectin [30,31]. In order to further characterize the reactivity of ACPA to citrullinated peptides, RA patient sera (*n* = 28), HC sera (*n* = 28) and MS sera (*n* = 10) were tested for reactivity to a citrullinated α-enolase peptide (KIHARCEIFDS-Cit-GNPTVEC) by ELISA.

As seen in Figure 1, elevated antibody reactivity was found to the citrullinated peptides compared to the Arg-containing control peptide (*p* = 0.0073 for the cyclic and *p* = 0.0003 for the linear). No significant difference in antibody reactivity was found between the cyclic and the linear α-enolase peptides (*p* = 0.2647). Approximately 40% of the RA sera recognized the α-enolase peptides, and reacted significantly to the linear and the cyclic peptide compared to the control peptide (*p* < 0.0001). None of HC sera or MS sera reacted to the citrullinated α-enolase peptides, confirming that ACPA reactivity to the α-enolase peptides was specific for RA.

### 3.2. Reactivity of Rheumatoid Arthritis Sera to Truncated Linear and Cyclic α-Enolase Peptides

Previous studies describing RA sera reactivity to citrullinated pro-filaggrin peptides indicated that ACPA reactivity is dependent on peptide length and conformation [23]. To determine whether this relates to α-enolase peptides as well, RA sera that were positive for reactivity to α-enolase in the preliminary screening (*n* = 12) were tested for reactivity to cyclic and linear truncated α-enolase peptides by ELISA.

As seen in Figure 2, RA sera recognized all of the linear peptides independent of their length. Sensitivities of approximately 90% were found for all of the linear peptides. In contrast, specific ACPA reactivity was primarily found to the cyclic peptides C-19-Cit and C-10-Cit, obtaining sensitivities of approximately 90% as well. These findings indicate that the peptide conformation affects the antibody reactivity and that it is essential for peptide presentation. Additionally, no specific reactivity was found when screening HC sera. These findings are in accordance with the literature, describing that ACPA reactivity to 19-mer linear and cyclic pro-filaggrin peptides yields similar reactivity, whereas ACPA reactivity to the linear citrullinated peptides is favoured for smaller peptides (<12 amino acids), when compared to the cyclic peptides [23].

### 3.3. Overlapping Reactivities of Anti-Citrullinated Protein Antibody Responses

As presented, approximately 40% of the RA sera reacted with the α-enolase peptides. To determine whether the ACPA reactivities were specific for the α-enolase peptide, all RA sera (*n* = 28) and HC sera (*n* = 28) were tested for reactivity to various citrullinated peptides in ELISA. Peptides from pro-filaggrin, proteoglycan, fibronectin, and EBNA-2 were tested for antibody reactivity.

As shown in Figure 3, significant RA antibody reactivity was found to the citrullinated pro-filaggrin (*p* = 0.0119), proteoglycan (*p* = 0.0004), fibronectin (*p* = 0.0459) and EBNA-2 (*p* < 0.0001) peptides compared to the HC. The proteoglycan and EBNA-2 peptides obtained the highest sensitivities of 50% and 68%, respectively. One HC serum showed low reactivity to EBNA-2 L (Figure 3b).

Thorough analysis of the reactivities of the RA sera showed that approximately 14% of the RA sera samples recognized all 4 peptides, whereas 18% reacted to 3 peptides. Note that approximately 54% of the samples recognized 1 or 2 peptides, whereas 14% of the RA cohort did not show reactivity to any of the peptides (Table 2, first column).

Of the 9 RA sera that only reacted with 1 peptide (Table 2, first column), 22% reacted with the proteoglycan peptide (*n* = 2), 22% with the fibronectin peptide (*n* = 2) and 56% (*n* = 5) with EBNA-2 peptide.

When dividing the complete RA cohort into α-enolase-positive and -negative sera, it was observed that the RA samples in the α-enolase-positive cohort were prone to have a higher degree of overlapping antibody reactivities compared to the α-enolase-negative cohort. For instance, 50% of the RA sera in the α-enolase-positive cohort recognized 3 or 4 peptides compared to 19% in the α-enolase-negative cohort. Similarly, 42% of the α-enolase-positive cohort recognized 1 or 2 peptides compared to 63% for the α-enolase-negative cohort.

When examining the reactivity of the 3 cohorts (complete cohort, α-enolase-positive cohort, α-enolase-negative cohort) to the three most reactive peptides, pro-filaggrin, proteoglycan, and EBNA-2 L, similar results were obtained. As presented in Figure 4, significant overlapping reactivities were found between the 3 cohorts.

Sera that only reacted to one peptide primarily recognized EBNA-2, as described earlier. Moreover, the most significant overlap in antibody reactivity was found between EBNA-2 and proteoglycan (Figure 4a,b), which is in accordance with the fact that these peptides obtained the highest sensitivities, as presented in Figure 3.

### 3.4. Reactivity of RA Sera to Linear and Cyclic Peptide Versions. Is the Effect of Cyclization on Antibody Sensitivity General?

Previous findings indicated that the cyclic version of the α-enolase peptide had a higher sensitivity, although not statistically significant when compared to the linear peptide. To determine whether this effect is general or peptide-specific, RA reactivity to linear and cyclic citrullinated pro-filaggrin and EBNA-2 peptides were tested in ELISA.

Figure 5 illustrates the reactivity of RA and HC sera to the linear and cyclic Pro-filaggrin and EBNA-2 peptide. ACPA reactivity to the citrullinated peptides was significantly elevated compared to the HCs (*p* = 0.0044 for EBNA-2 L, *p* < 0.0001 for EBNA-2 C, *p* = 0.0028 for pro-filaggrin L, *p* < 0.0001 for pro-filaggrin C).

Moreover, cyclization was observed to increase assay sensitivity, as the cyclic EBNA-2 and pro-filaggrin peptides had higher sensitivitities compared to the linear peptides (*p* = 0.0331), 67% and 44% of the RA sera reacted to the cyclic and linear EBNA-2 peptide, respectively, whereas 68% and 44% of the RA sera reacted to the cyclic and linear pro-filaggrin peptides, respectively. No statistically significant difference in antibody reactivity to the linear and cyclic pro-filaggrin peptides was found, although the antibody reactivity to the cyclic version was elevated (Figure 5b).

Collectively, these findings indicate that peptide conformation and peptide length affect antibody reactivity.

## 4. Discussion

In the present study, we analysed the reactivity of peptide-specific ACPAs to citrullinated epitopes and confirmed that factors such as peptide length and conformation notably influence antigen presentation.

The sensitivity of the full-length α-enolase peptide was determined to be approximately 44%, which is supported by earlier findings described in the literature [32]. ACPA reactivity to α-enolase was originally described by Lundberg et al., who showed that a cyclic peptide obtained the highest antibody reactivity among the human α-enolase and *Porphyromonas gingivalis* enolase peptides tested [32]. Similarly, no reactivity was found to the Arg-containing control peptide, confirming that the ACPAs are citrulline-specific (Figure 1). In addition to this, MS samples were tested for reactivity to the linear α-enolase peptide, as it has been described that citrullinated protein levels are elevated in MS [33], however, no reactivity was observed, confirming that ACPA reactivity is specific for RA.

The above mentioned experiments using α-enolase peptides were conducted in the absence of reducing agents, thus the linear peptides may in theory be able to cyclize during coating; however, as the effect of cyclization for α-enolase peptides was the same for pro-filaggrin and EBNA-2 peptides, where no cysteines were found in the linear peptides, we have reason to believe that the α-enolase peptides were found in a linear form during coating [23].

Screening of the truncated α-enolase peptides showed that all of the linear peptides (L-19-Cit, L-14-Cit, L-12-Cit and L-10-Cit) were roughly recognized to the same extent by the RA cohort, indicating that the length of the linear peptides is less important compared to the cyclic peptides. These findings confirm that the Cit–Gly motif in combination with a peptide backbone of approximately 10 amino acids, perhaps even shorter, is sufficient for antibody binding, as previously proposed [23]. In terms of reactivity, L-14-Cit, which is the second-longest peptide, showed the highest reactivity. In a study conducted using pro-filaggrin peptides, it was found that a linear 21-mer peptide and a linear 14-mer peptide were significantly recognized by ACPAs. Moreover, the 14-mer peptide obtained the highest sensitivity, which conforms to this study, favouring peptides of approximately 10–14 amino acids [27]. This effect may relate to the peptide structure. Even though a small number of amino acids are flanking the Cit–Gly motif in the shorter peptides, the peptides are still able to fold and thus acquire a specific conformation that appears to bind to ACPAs more efficiently. The exact reason remains to be determined.

Increased sensitivities were obtained for the longest (C-19-Cit) and the shortest (C-10-Cit) cyclic α-enolase peptides compared to the C-14-Cit and C-12-Cit peptides, indicating that peptide length and conformation are crucial for peptides containing 12–19 amino acids. These findings are supported by the literature, where it has been reported that the reduced number of amino acids in the cyclic structure may constrain the peptide in a more locked conformation, reducing the flexibility of the peptides and hence negatively influencing the ACPA reactivity [16]. The fact that the smaller peptide (C-10-Cit) was as sensitive as the longest peptide (C-19-Cit), and thus more sensitive than the mid-length peptides (C-12-Cit and C-14-Cit), is intriguing. This interesting reactivity pattern to truncated cyclic peptides could be further investigated by performing crystallography studies of the ACPA binding groove. Ultimately, these findings regarding truncated peptides (both linear and cyclic) confirm that peptide length and conformation are essential for antibody reactivity.

Concerning the aspect of cyclization, it has been reported that peptide cyclization has a positive effect upon antibody reactivity [14]. Our studies of pro-filaggrin and EBNA-2 cyclic and linear peptides revealed an evident increase in the antibody reactivity to the cyclic peptides compared to the linear peptides. These results are consistent with the longest α-enolase peptides (L-19-Cit and C-19-Cit). Nevertheless, for the α-enolase peptides, the reactivity to linear peptides did not depend on peptide length, whereas ACPA reactivity to cyclic peptides was length-dependent. This effect is in direct contrast to earlier findings using cyclic and linear pro-filaggrin peptides, where ACPA reactivity to both the linear and the cyclic peptides appeared to be length-dependent [27]. This may in theory be ascribed to the peptides used, as the pro-filaggrin and EBNA-2 peptides were biotinylated, whereas the α-enolase peptides had free terminals, hence the absence of biotin may have influenced the peptide coating. Thus, further analyses are necessary to confirm these results.

When focusing on the peptide sequence, a high degree of homology is found between pro-filaggrin and EBNA-2 in the C-terminal region, where positively charged and small amino acids are present (Table 1). The high degree of sequence homology may explain the similar sensitivities that the peptides yielded, suggesting that peptide sequence influences antibody reactivity. However, the Proteoglycan Cit peptide sequence does not have homology in the C-terminal end to pro-filaggrin and EBNA-2 and still yields a sensitivity of approximately 50%. Since several sequence patterns along with a Cit–Gly motif can be found among the peptides tested, these observations led to the hypothesis that a structural homology could be shared by the citrullinated peptides recognized by ACPAs. This remains to be elaborated.

Screening of the RA and the HC cohorts on a peptide panel, including four citrullinated peptides from pro-filaggrin, proteoglycan, fibronectin and EBNA-2, revealed a significantly different reactivity between RA and HC samples (Figure 3). The citrullinated peptides obtained the following sensitivities: Pro-filaggrin 32%, Fibronectin 36%, Proteoglycan 50% and EBNA-2 68%. In addition, more than 50% of the RA samples reacted with more than one peptide of the panel, confirming the ability of ACPAs to bind to several citrullinated targets. Further investigation analysing the overlapping reactivities of the RA sera within the peptide panel (Figure 3) showed that approximately 32% reacted with 3 or 4 peptides, 21% with 2 peptides, 14% did not react at all and, lastly, 32% of the cohort only showed reactivity to one peptide, which in 56% of the cases was the EBNA-2 peptide, suggesting the presence of EBNA-2-specific ACPAs. The fact that most of the RA samples interacted with more than one peptide supports the theory of the overlapping reactivity of ACPAs, highlighting the central role of the Cit–Gly motif together with the surrounding amino acids for antigen–antibody binding.

The previous findings led us to compare the ACPA reactivities between the peptide panel and the α-enolase peptides, dividing them into α-enolase-positive and -negative ones. As shown in Table 2, within the α-enolase-positive cohort (*n* = 12), all the RA sera reacted with EBNA-2 peptide and no monospecific reactivity was detected to pro-filaggrin and proteoglycan peptides. One RA sample of the α-enolase-positive cohort only showed reactivity to the α-enolase peptide and not to the whole peptide panel, indicating that it was specific for the side-chains of the α-enolase epitope rather than the actual backbone in combination with a central Cit–Gly motif. In the α-enolase-negative cohort, more monospecific reactivities were observed. These findings are in accordance with the literature, which reports that approximately 15% of the RA sera are monospecific [30]. Additionally, although a small part of the RA sera reacted with only one citrullinated peptide, these results confirm the theory of overlapping and nonoverlapping ACPA reactivities. Here, the α-enolase-positive cohort was regarded to have overlapping ACPAs, which are considered as backbone-dependent, whereas the α-enolase-negative cohort was accounted to the nonoverlapping group of ACPAs, which depend on the flanking amino acid side-chains to establish a stable antigen–antibody interaction [16]. Collectively, the RA sera that were positive for α-enolase turned out to be more overlapping within the peptide panel compared to the α-enolase-negative cohort.

## 5. Conclusions

RA is an autoimmune disease that affects many people and reduces their quality of life. Thus, early diagnosis of the disease is fundamental to undertake therapy as soon as possible and to prevent disease progression. One of the most important diagnostic criteria is the detection of autoantibodies directed to a variety of citrullinated antigens in the serum of the patient. The origin of these autoantibodies is still unknown, and thus, gaining knowledge about the epitopes that ACPAs are able to recognize is important in the development of more sensitive and more specific diagnostic tools and to obtain a better understanding of the pathophysiological mechanisms. For these purposes and to elucidate the etiology of RA, interactions between ACPAs and potential candidate (auto)antigens were analysed.

Regarding the experiments to further investigate the structure and composition of ACPAs, these results confirm their cross-reactive nature. The analysed RA sera showed a different reactivity pattern to the citrullinated peptide panel in relation to the α-enolase-positive and -negative cohorts. This result indicates that ACPAs can be divided into two categories based on their ability to bind to a wider or a more limited number of citrullinated targets, which, respectively, reflects the peptide backbone and peptide side-chains dependencies. Especially, the ability of ACPAs to react with various citrullinated targets can be explained by a similar structure of citrullinated epitopes. Early studies showed that positively charged and small amino acids in the C-terminal end relative to citrulline yield high sensitivities. However, not all the peptides tested in this study have these features but can still interact with ACPAs (Table 1). Therefore, a structural homology, rather than sequence homology, should be important for ACPA recognition of the citrullinated targets, and this would, in addition, support the theory that the overlapping group of ACPAs is backbone-dependent. To obtain a better knowledge of the structure and to find a pattern that brings together all the citrullinated targets that interact with ACPAs, circular dichroism analyses could be performed.

Within the peptides tested on the RA cohort, the epitope originating from α-enolase, a known autoantigen in RA, was confirmed to have a lower sensitivity compared to available commercial assays and other citrullinated peptides, such as the EBNA-2 peptide, which was confirmed to be a highly sensitive substrate, even though contribution of EBV infection to RA onset still needs to be clarified.

Collectively, this study contributes to the understanding of the nature of ACPAs.

## Figures and Tables

**Figure 1 antibodies-10-00027-f001:**
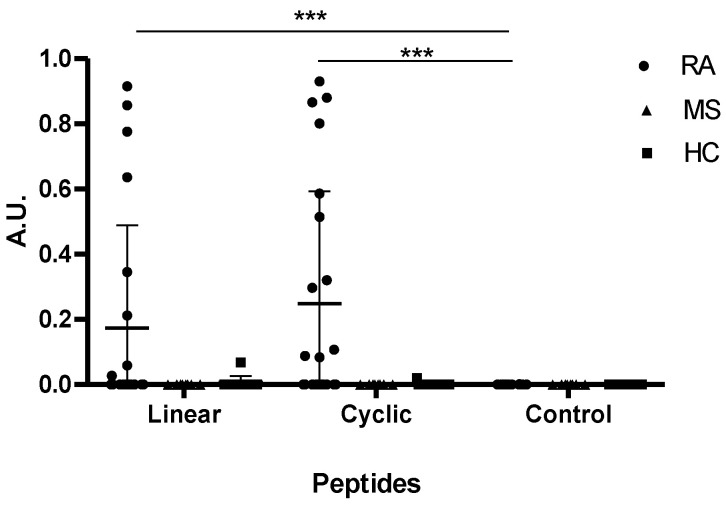
Reactivity of rheumatoid arthritis (RA), healthy control (HC) and multiple sclerosis (MS) sera to α-enolase peptides (KIHARCEIFDS-Cit-GNPTVEC) analysed by traditional ELISA. A linear Arg-containing peptide (KIHARCEIFDS-R-GNPTVEC) was used as negative control. HC and MS sera were used as controls. A.U. were defined as absorbances normalized relative to a positive RA control pool. *p* values less than 0.001 are shown as ***.

**Figure 2 antibodies-10-00027-f002:**
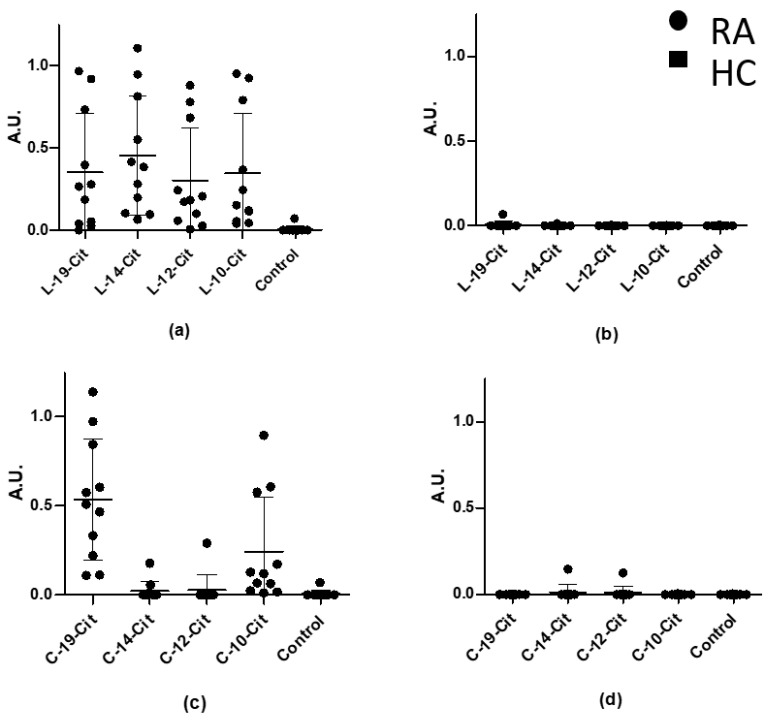
Reactivity of rheumatoid arthritis (RA) sera and healthy control (HC) sera samples to truncated linear and cyclic citrullinated α-enolase peptides analysed in ELISA. Peptides are plotted from left to right with decreasing length. An Arg-containing peptide was used as control (KIHARCEIFDSRGNPTVEC). (**a**) Reactivity of RA sera to linear peptides. (**b**) Reactivity of HC sera to linear peptides. (**c**) Reactivity of RA sera to cyclic peptides. (**d**) Reactivity of HC sera to cyclic peptides. A.U. were defined as absorbances normalized relative to a positive RA control pool.

**Figure 3 antibodies-10-00027-f003:**
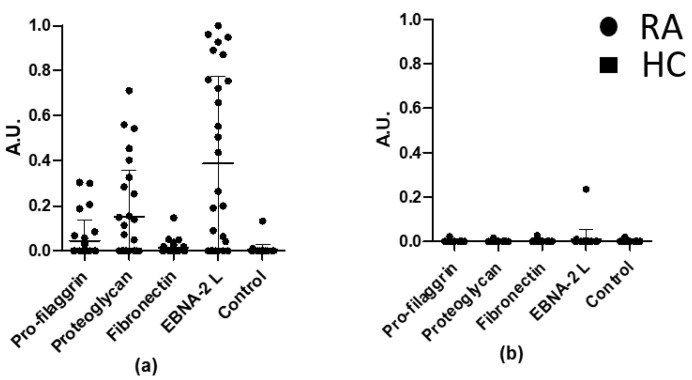
Reactivity of rheumatoid arthritis (RA) RA and healthy control (HC) sera to a selected peptide panel. (**a**) Screening of RA samples (*n* = 28) to citrullinated peptides from pro-filaggrin, proteoglycan, fibronectin and EBNA-2. An Arg-containing pro-filaggrin peptide was used as control. (**b**) Screening of HC sera (*n* = 28) to citrullinated peptides from pro-filaggrin, proteoglycan, fibronectin and EBNA2. An Arg-containing pro-filaggrin peptide was used as control. A.U. were defined as absorbances normalized relative to a positive RA control pool.

**Figure 4 antibodies-10-00027-f004:**
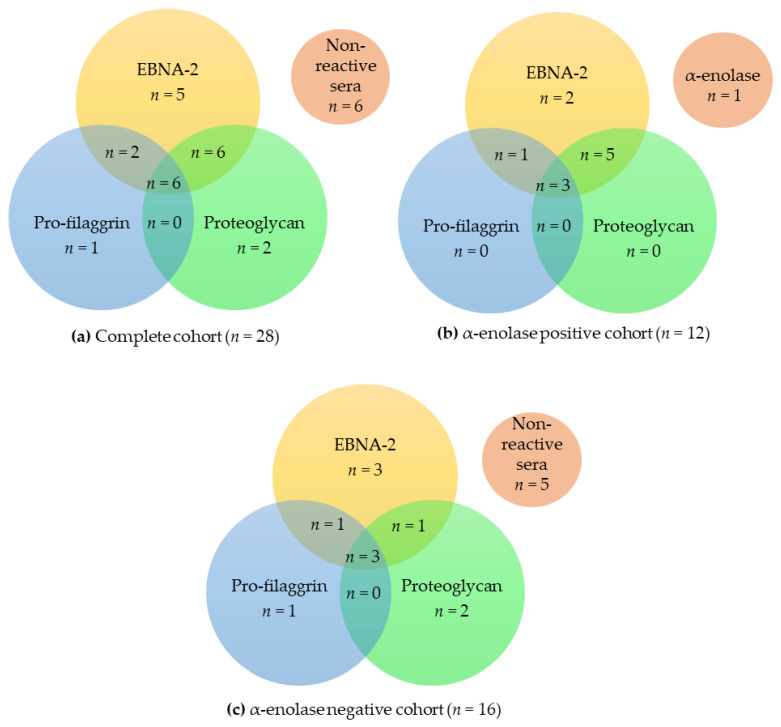
Venn diagram illustrating overlapping anti-citrullinated protein antibody reactivities to EBNA-2, pro-filaggrin and proteoglycan. (**a**) Reactivities of the complete rheumatoid arthritis (RA) cohort to the selected peptides (*n* = 28). (**b**) Reactivities of α-enolase-positive RA serum samples (*n* = 12). (**c**) Reactivities of α-enolase-negative RA serum samples (*n* = 16).

**Figure 5 antibodies-10-00027-f005:**
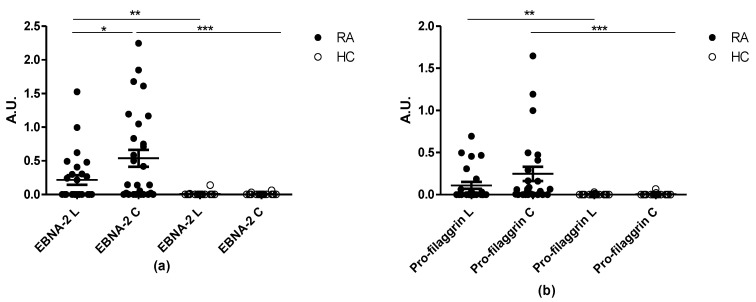
Reactivity of rheumatoid arthritis (RA) samples and healthy control (HC) samples to pro-filaggrin (25 RA and 25 HC samples) and EBNA-2 (28 RA and 28 HC samples) peptides in their linear and cyclic conformation. (**a**) Reactivity of RA and HC sera to citrullinated cyclic and linear EBNA-2 peptides. (**b**) Reactivity of RA and HC sera to citrullinated cyclic and linear pro-filaggrin peptides. A.U. were defined as absorbances normalized relative to a positive RA control pool. *** *p* < 0.001, ** *p* < 0.01, * *p* < 0.05.

**Table 1 antibodies-10-00027-t001:** Synthetic peptides tested for antibody reactivity. “B” represents biotin.

Origin	Name	Sequence
α-enolase	C-10-Cit	CFDS-Cit-GNPTC
	L-10-Cit	CFDS-Cit-GNPTC
	C-12-Cit	CIFDS-Cit-GNPTVC
	L-12-Cit	CIFDS-Cit-GNPTVC
	C-14-Cit	CEIFDS-Cit-GNPTVEC
	L-14-Cit	CEIFDS-Cit-GNPTVEC
	C-19-Cit	KIHARCEIFDS-Cit-GNPTVEC
	L-19-Cit	KIHARCEIFDS-Cit-GNPTVEC
	L-19-Arg	KIHARCEIFDSRGNPTVEC
Fibronectin	Fibronectin L	DHEGTHSTK-Cit-GHAKSRPVRD(K(B))
Proteoglycan	Proteoglycan L	B-PQASVPLRLT-Cit-GSRAPISRAQ
Pro-filaggrin	Pro-filaggrin C	HQCHQEST-Cit-GRSRGRCGRSGS(K(B))
	Pro-filaggrin L	HQSHQEST-Cit-GRSRGRSGRSGS(K(B))
Epstein–Barr virus	EBNA-2-L	GQGRGRWRG-Cit-GRSKGRGRMH(K(B))
Nuclear antigen 2	EBNA-2-C	GQGRCGRWRG-Cit-GRSKGRGCRMH(K(B))

**Table 2 antibodies-10-00027-t002:** Reactivities of the RA cohort to citrullinated peptides originating from pro-filaggrin, proteoglycan, fibronectin and EBNA-2.

	Total RA Cohort	α-Enolase Positive RA Cohort	α-Enolase Negative RA Cohort
*n*	28	12	16
4 reactive peptides	4 (14.3%)	2 (16.7%)	2 (12.5%)
3 reactive peptides	5 (17.9%)	4 (33.3%)	1 (6.3%)
2 reactive peptides	6 (21.4%)	3 (25.0%)	3 (18.8%)
1 reactive peptide	9 (32.1%)	2 (16.7%)	7 (43.8%)
0 reactive peptides	4 (14.3%)	1 (8.3%)	3 (18.8%)

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
