# Peer review of "Specificity of Anti-Citrullinated Protein Antibodies to Citrullinated α-Enolase Peptides as a Function of Epitope Structure and Composition"

_2073-4468, 2021, doi:10.3390/antib10030027_

Round 1

Reviewer 1 Report

  1. The overall findings had been reported before, and the data were supported by other literature. It seems just another supporting document without novelty.
  2. Line 5, no affiliation of authors #4.
  3. Line 21, remove peptide (duplicated)
  4. Lines 59-61. Too many which clauses, please revise.
  5. Line 68, it's not clear what does it mean "unknow for proprietary reasons."
  6. Line 117, for Table 1, suggest to include "C" for cyclic and "L" for linear peptides.
  7. Line 120, suggest to have capital B for biobank.
  8. Line 121, analyses should be a verb "analyzes" here.
  9. Line 132, Please have the abbreviation TTN described when the buffer was introduced in Line 110-111.
  10. Line 135, please describe pNPP.
  11. Line 139, same as #9 for CB.
  12. Line 153, the total HC sera number was 28 (Line 120), why is 22 here?
  13. The statistical analysis for Figure 1 included those sera had no reactions with the peptides. Why does it make sense?
  14. How to define "reactive?" As long as there is absorbance observed from ELISA assay? What was the assay variability? 
  15. Line 181-182, the authors stated "whereas the reactivity to the linear conformation of citrullinated peptides is favored for smaller peptide." But from Figure 2(a), no such favor observed.
  16. Line 198-199, the authors stated "None of the HC sera recognized the various citrullinated peptides (Figure 3b). But there was at least one serum reacted to EBNA-2 L peptide.
  17. Line 212, there is not Table 2a. Please change it to the "Table 2, first column). 
  18. Line 244, typo "andd"
  19. Line 290, please start a new sentence from "moreover." 
  20. The truly linear peptides in this study are questionable, as the authors stated in discussion. It is hard to believe that the "linear" peptides won't form disulfide bond in the alkaline conditions. Although Figure 2a and 2c did show differences between the "linear" and the cyclic peptides, that cannot support those "linear" peptides are truly linear. Authors should have other lines of evidence to demonstrate the structures of these "linear" peptides.

Author Response

Dear Editor,

Thank you for the constructive comments raised by the two reviewers, which have improved the manuscript. We have addressed all comments and concerns raised. Please find our response to the individual comments below.

Best regards

Gunnar Houen and Nicole Trier

  1. The overall findings had been reported before, and the data were supported by other literature. It seems just another supporting document without novelty.

Response: we agree that ACPAs in general have been have been described very well in the existing literature. However, the reactivity of ACPA to systematically truncated enolase peptides is new and may contribute to determine what actually is necessary for a stable interaction. Moreover, this study contributes with novelty in relation to optimal epitope presentation, as the effect of peptide cyclization only has been very sporadically described in the literature and with only very limited peptide examples. 

  1. Line 5, no affiliation of authors #4.

Response:Corrected

  1. Line 21, remove peptide (duplicated)

Response:Corrected

  1. Lines 59-61. Too many which clauses, please revise.

Response: Amended as requested, the paragraph has been rephrased.

  1. Line 68, it's not clear what does it mean "unknow for proprietary reasons."

Response: The sentence has been rephrased and that specific part has been deleted.

  1. Line 117, for Table 1, suggest to include "C" for cyclic and "L" for linear peptides.

Response: Amended as requested.

  1. Line 120, suggest to have capital B for biobank.

Response:Corrected

  1. Line 121, analyses should be a verb "analyzes" here.

Response:Corrected

  1. Line 132, Please have the abbreviation TTN described when the buffer was introduced in Line 110-111.

Response:Corrected

  1. Line 135, please describe pNPP.

Response:Corrected

  1. Line 139, same as #9 for CB.

Response:Corrected

  1. Line 153, the total HC sera number was 28 (Line 120), why is 22 here?

Response: Corrected, a typo.

  1. The statistical analysis for Figure 1 included those sera had no reactions with the peptides. Why does it make sense?

Response: The statistical analysis of the reactive RA sera has been added.

  1. How to define "reactive?" As long as there is absorbance observed from ELISA assay? What was the assay variability?

Response: The definition of a reactive sample and assay variability has been elaborated in the materials and methods section. 

  1. Line 181-182, the authors stated "whereas the reactivity to the linear conformation of citrullinated peptides is favored for smaller peptide." But from Figure 2(a), no such favor observed.

Response: The sentence has been rephrased.

  1. Line 198-199, the authors stated "None of the HC sera recognized the various citrullinated peptides (Figure 3b). But there was at least one serum reacted to EBNA-2 L peptide.

Response: The sentence has been rephrased.

  1. Line 212, there is not Table 2a. Please change it to the "Table 2, first column). 

Response: Amended as requested.

  1. Line 244, typo "andd"

Response: Corrected

  1. Line 290, please start a new sentence from "moreover." 

Response: Amended as requested

  1. The truly linear peptides in this study are questionable, as the authors stated in discussion. It is hard to believe that the "linear" peptides won't form disulfide bond in the alkaline conditions. Although Figure 2a and 2c did show differences between the "linear" and the cyclic peptides, that cannot support those "linear" peptides are truly linear. Authors should have other lines of evidence to demonstrate the structures of these "linear" peptides.

Response: As seen in table 1, only the linear peptides from α-enolase contained cysteines. The linear peptides from EBNA2 and pro-filaggrin do not contain cysteines and cannot cyclize. The effect of cyclization for α-enolase is similar to EBNA2 and pro-filaggrin, thus we believe that the results support that the peptides are not cyclized, although we cannot rule out that some degree of cyclization may occur. The linear form of the peptides was confirmed by mass spectrometry and adsorption to the solid phase polystyrene surface (coating) is believed to be quite fast, although for convenience, it is carried out overnight. In any case, the results show that there is substantial difference in the reactivity of the cyclic and linear peptide preparations, thus supporting the conclusions.

Reviewer 2 Report

The topic of rheumatoid arthritis diagnosis is extremely important
and in fact there is no perfect diagnostic method to date. It is
believed that ACPAs are currently one of the best biomarker of the
disease, unfortunately not ideal, so research on improving their
prediction power is extremely important.

I really like the idea of ​​studying cyclic and linear peptides
and the normalization based RA pool.

  1.   I don't understand why in analysis of reactivity of RA, HC and MS sera
    to α-enolase peptides, pooled HC sera were used as controls not individual sera. This type of approach may distort the obtained results. Do you have results for individual HD sera?
  2. In chapter Cross-reactivities of anti-citrullinated protein antibody responses, talking about a cross-reaction is an overinterpretation, in the tested system we can only talk about reactivity, unless the reactions were conducted on a pure system.   

Author Response

Dear Editor,

Thank you for the constructive comments raised by the two reviewers, which have improved the manuscript. We have addressed all comments and concerns raised. Please find our response to the individual comments below.

Best regards

Gunnar Houen and Nicole Trier

Reviewer 2:

  1.   I don't understand why in analysis of reactivity of RA, HC and MS sera
    to α-enolase peptides, pooled HC sera were used as controls not individual sera. This type of approach may distort the obtained results. Do you have results for individual HD sera?

Response: Amended as requested. The HC pool has been replaced with individual serum samples.

  1. In chapter Cross-reactivities of anti-citrullinated protein antibody responses, talking about a cross-reaction is an overinterpretation, in the tested system we can only talk about reactivity, unless the reactions were conducted on a pure system.   

Response: We agree that the term cross-reactivity is an over-interpretation of the presented results. Paragraphs concerning this topic have been rephrased. 

Round 2

Reviewer 1 Report

no further comments.

Author Response

Dear reviewer.

Moderate English changes has been made. 

Reviewer 2 Report

Thank you very much for taking into account my comments.
Please improve Figure 1, where it remains HC pool.

Author Response

Dear reviewer.

Thank you for your comment. Figure 1 has been corrected.